# SGAEMDA: Predicting miRNA-Disease Associations Based on Stacked Graph Autoencoder

**DOI:** 10.3390/cells11243984

**Published:** 2022-12-09

**Authors:** Shudong Wang, Boyang Lin, Yuanyuan Zhang, Sibo Qiao, Fuyu Wang, Wenhao Wu, Chuanru Ren

**Affiliations:** 1College of Computer Science and Technology, Qingdao Institute of Software, China University of Petroleum, Qingdao 266580, China; 2School of Information and Control Engineering, Qingdao University of Technology, Qingdao 266525, China

**Keywords:** miRNA, disease, association prediction, stacked graph autoencoder, higher-order features

## Abstract

MicroRNA (miRNA)-disease association (MDA) prediction is critical for disease prevention, diagnosis, and treatment. Traditional MDA wet experiments, on the other hand, are inefficient and costly.Therefore, we proposed a multi-layer collaborative unsupervised training base model called SGAEMDA (Stacked Graph Autoencoder-Based Prediction of Potential miRNA-Disease Associations). First, from the original miRNA and disease data, we defined two types of initial features: similarity features and association features. Second, stacked graph autoencoder is then used to learn unsupervised low-dimensional representations of meaningful higher-order similarity features, and we concatenate the association features with the learned low-dimensional representations to obtain the final miRNA-disease pair features. Finally, we used a multilayer perceptron (MLP) to predict scores for unknown miRNA-disease associations. SGAEMDA achieved a mean area under the ROC curve of 0.9585 and 0.9516 in 5-fold and 10-fold cross-validation, which is significantly higher than the other baseline methods. Furthermore, case studies have shown that SGAEMDA can accurately predict candidate miRNAs for brain, breast, colon, and kidney neoplasms.

## 1. Introduction

MicroRNA (miRNA) is a single-stranded small molecule RNA with a length of about 19–25 nucleotides that is encoded by endogenous genes [1,2]. MiRNAs are linked to and play a crucial part in many vital human body processes, such as cell proliferation, differentiation, immunity, and metabolism [3]. As a result, miRNAs have received increased attention, particularly in the field of associations between miRNAs and complex human diseases. Overexpression and downregulation of miRNA expression in humans have been linked to a variety of complex diseases, according to research [4,5]. Upregulation of miR-17-5p expression, for example, has a greater effect on pancreatic cancer cell proliferation and significantly increases the number of invading cells [6]. When compared to normal breast tissue, abnormal expression of miRNAs such as mir-125b, mir-145, mir-21, and mir-155 causes human breast cancer [7]. Cressatti et al. [8] discovered that miR-153 and miR-223 could be used as biomarkers for Parkinson’s disease (PD) diagnosis through paired regulation of α-synuclein. MiR-34, miR-124a, -146, miR-187, miR-199a-5p, miR-203, miR-210, and miR-383 dysregulation all have a negative impact on pancreatic β-cell viability and function, which leads to uncontrolled proliferation of insulin-secreting cells and the development of diabetes [9,10]. In conclusion, miRNAs have been shown to be inextricably linked to the emergence of many human complex diseases, making the prediction of potential miRNA-disease association (MDA) a promising area of research. It can help researchers comprehend the pathological mechanisms of complex diseases, which can be beneficial in both the treatment and diagnosis of complex diseases.

Traditional biological wet experiments, such as anchored polymerase chain reaction and reverse transcription polymerase chain reaction, were used in the early years to identify the relationship between miRNAs and diseases, but they all have drawbacks such as complicated experiments, long time periods, and high costs [11,12,13]. Several studies in the field of bioinformatics have been developed in recent years, such as drug–drug interactions [14], drug–target interactions [15], lncRNA–disease association prediction [16], and lncRNA–miRNA interaction [17]. Each of these studies has added to our understanding of computational approaches for predicting miRNA–disease connections. Many superior computational methods for predicting potential miRNA–disease associations have been proposed as more biological data sets have been collected, which not only saves significant money and time but also provides researchers with a new perspective to further validate the predicted potential associations. These MDA prediction computational approaches can be roughly categorized into three categories [18]: machine learning-based prediction models, deep learning-based prediction models, and matrix transformation-based prediction models.

Machine learning has been widely applied in all areas, and numerous machine learning models for predicting MDA have produced positive results. As there are not enough known miRNA–disease connections, existing prediction models perform poorly, Zhou et al. [19] presented a new model combining gradient boosting decision tree and logistic regression (GBDT-LR) to rank miRNA candidates for diseases. The model can extract features and then score them using logistic regression. Peng et al. [20] proposed a new prediction model called Ensemble of Kernel Ridge Regression-based MiRNA-Disease Association prediction (EKRRMDA), which used KRR to build two classifiers in miRNA space and disease space, respectively, and combined them with ensemble learning to improve model prediction accuracy. Liu et al. [21] created a computational model for the SMALF by learning potential features from the original miRNA–disease association matrix and then predicting unknown miRNA–disease associations using XGBoost. Tang et al. [22] developed an ensemble learning method (PMDFI) based on higher-order feature interactions to predict potential miRNA–disease associations. It uses stacked autoencoders to learn higher-order features from the similarity matrix and then uses an integrated model combining multiple random forests with logistic regression to predict an association. Liu et al. [23] proposed an autoencoder-based deep forest ensemble learning model (DFELMDA), which was further validated through case studies of colon, breast, and lung tumors with varying disease types. Both PMDFI and DFELMDA use automatic encoders, but as they do not consider graph structure information, they cannot learn the miRNA and disease feature representation well. Although machine learning-based methods have demonstrated good performance, they typically require domain knowledge to build sample features.

With the advent of Deep Learning, many methods of end-to-end computing have been developed, and this novel prediction method predicts better than earlier traditional machine learning methods. Xuan et al. [24] developed CNNMDA, a deep learning method that uses two convolutional neural networks to efficiently learn the potential relationship between miRNAs and diseases (CNN). Li et al. [25] created a GAEMDA model that takes miRNA and disease similarity as feature information, aggregates it using a graph neural network-based encoder to generate a low-dimensional representation of the nodes, and finally predicts it using a bilinear decoder. Zhou et al. [26] proposed a deep self-coding multicore learning approach (DAEMKL) the following year, which uses multicore learning to build miRNA-disease heterogeneous networks and then uses regression models to learn their feature representations. Li et al. [27] designed a computational framework based on graph attention network fusion of multi-source information (GATMDA). It utilized the graph attention network to aggregate information from neighbors with different weights to extract nonlinear features of diseases and miRNAs, and then predicted MDA by efficiently fusing linear and nonlinear features of diseases and miRNAs through a random forest algorithm. Han et al. [28] proposed that LAGCN build a heterogeneous network by integrating miRNA similarity, disease similarity, and miRNA-disease association information, and then use the attention mechanism to synthesize multiple CNNs to learn miRNA and disease embedding. Although deep learning-based methods can learn feature representations automatically and improve model prediction performance to some extent, they require a large number of training samples and do not incorporate graph structure information, making it difficult to capture neighborhood information in the network.

Furthermore, in recent years, several MDA prediction algorithms based on matrix transformation have appeared. Yu et al. [29] proposed a prediction model based on matrix completion and label propagation (MCLPMDA). It used matrix completion to reconstruct a new miRNA and disease similarity matrix based on the miRNA-disease association matrix, and then used the label propagation algorithm to predict MDA. Gao et al. [30] proposed the Nearest Profile-based Collaborative Matrix Factorization (NPCMF) algorithm, which uses L2,1-norm to complete the unknown association, using miRNA and disease nearest neighbor information to construct similarity functions and thus find new MDAs. Chen et al. [31] proposed the neighborhood constraint matrix completion algorithm (NCMCMDA), which combined neighborhood constraints with matrix completion for assisted prediction before transforming the prediction task into an optimization problem that could be solved by a rapid iterative algorithm. Yin et al. [32] created a new computational model called Logistic Weighted Profile-based Collaborative Matrix Factorization by combining two methods, weighted profile and collaborative matrix factorization (LWPCMF). The findings show that LWPCMF can accurately predict potential MDA. Although the matrix transformation-based method overcomes the problem of feature representation using vectors in high-dimensional space, its results are highly dependent on the initial solution selection, and it often fails to converge, which is time-consuming.

Although the models presented above predicted MDA well, they do have certain limitations. In recent years, autoencoders have been widely used in various fields [33,34] to efficiently learn the feature representation of miRNAs and diseases without losing the graph structure topology information, we propose a stacked graph autoencoder-based miRNA-disease association prediction algorithm (SGAEMDA), as shown in Figure 1. All miRNA features were then concatenated with disease features as miRNA-disease pair features. We employed 5-fold and 10-fold cross-validation to evaluate the prediction performance of our method. As a consequence, the AUCs of SGAEMDA in 5-fold and 10-fold cross-validation were 0.9585 and 0.9616, respectively, much higher than the other baseline methods. In addition, to demonstrate SGAEMDA’s performance, we conducted case studies on brain neoplasms, breast neoplasms, colon neoplasms, and kidney neoplasms. According to the findings, the bulk of our predicted possible miRNA-disease associations were verified by the dbDEMC and miRCancer databases. This paper’s significant contributions are summarized as follows.

(1)We integrated both association information and similarity information to construct the initial features and could better learn the potential information in miRNA-disease pairs.(2)We propose a stacked graph autoencoder prediction framework. Unlike previous stacked autoencoders, which used layer-by-layer training, the stacked graph autoencoder uses multi-layer collaborative unsupervised training. It is capable of effectively extracting potential, deep, and unknown feature information from the similarity network to compensate for the shortcomings of previous models’ prediction results, which are biased toward miRNAs and diseases with known associations.(3)We use a multilayer perceptron (MLP) for prediction of the final results, which has high fault tolerance and can learn feature information from miRNA-disease pairs rapidly and efficiently to improve model prediction performance.

**Figure 1 cells-11-03984-f001:**
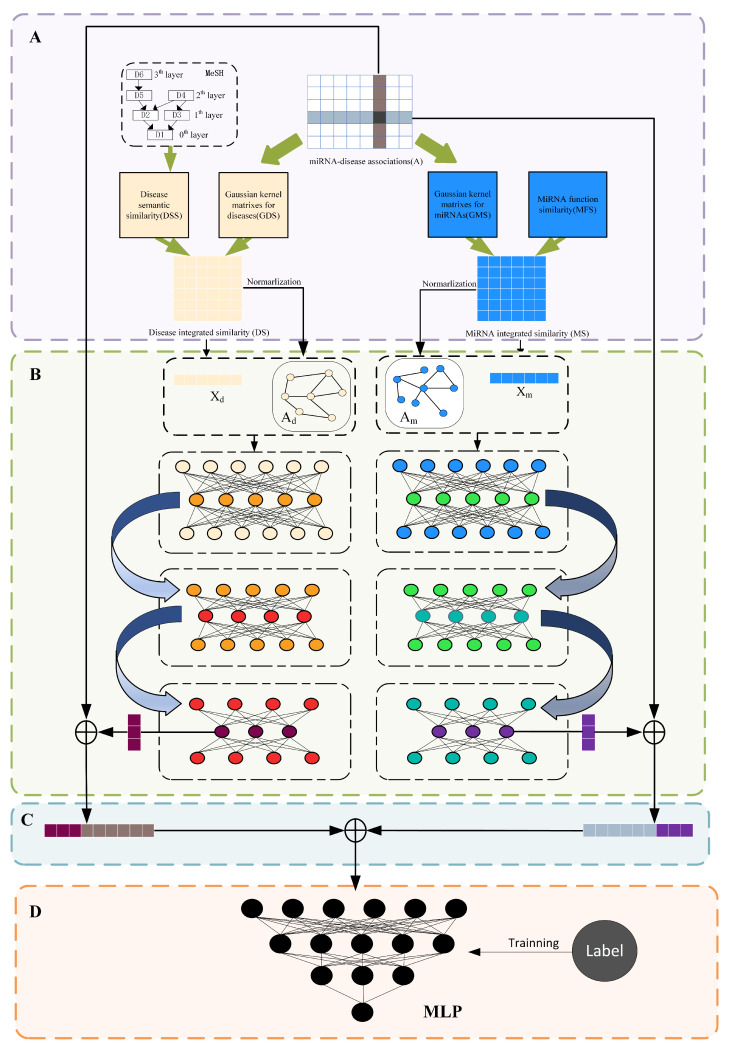
SGAEMDA flowchart. (**A**) Construction of initial features and data processing. (**B**) Pre-training to extract low-dimensional similarity features of miRNA and disease. (**C**) Fusion of learned miRNA and disease features to generate miRNA-disease pair feature vector. (**D**) Association prediction score by MLP.

## 2. Materials and Methods

### 2.1. Datasets for MDA Prediction

The Human miRNA-disease association dataset we used was downloaded from the HMDDv2.0 database [35]. It contains 5430 known associations of 383 complex diseases and 495 miRNAs, and the rest are unknown associations. In the follow-up experiments, we used a binary adjacency matrix A with $n_{m}$ rows and $n_{d}$ columns to storage all known and unknown associations. Where $n_{m}$ and $n_{d}$ are the number of miRNAs and diseases in this dataset, respectively. Specifically, this binary association matrix A is defined as follows:
(1)$$A\left(i,j\right)=\left\{\begin{array}{cc} 1, & \;\mathrm{if}\;\mathrm{miRNA}\;m_{i}\;\mathrm{is}\;\mathrm{associated}\;\mathrm{to}\;\mathrm{disease}\;d_{j}\\ 0, & \;\mathrm{otherwise}\; \end{array}\right.,$$

### 2.2. MiRNA and Disease Informaton

#### 2.2.1. MiRNA Function Similarity

Wang et al. [36] proposed a method to measure miRNA functional similarity and a method to construct miRNA functional similarity networks based on the hypothesis that functionally similar miRNAs are often associated with similar diseases. The functional similarity information of miRNAs can be obtained from http://www.cuilab.cn/files/images/cuilab/misim.zip (accessed on 23 May 2022). Then, based on the obtained information, we built the miRNA functional similarity matrix MFS with $n_{d}$ rows and $n_{d}$ columns. Where $MFS(m_{i},m_{j})$ denotes the functional similarity score between miRNA $m_{i}$ and miRNA $m_{j}$.

#### 2.2.2. Disease Semantic Similarity

Based on a previous study [37], disease semantic similarity was obtained based on statistics disease ontology information. Specifically, all disease semantic similarities can be calculated using medical subject headings (MeSH), where each disease $d_{i}$ can be described by several directed acyclic graphs (DAGs). The directed acyclic graph can be defined as $DAG\left(d_{i}\right)=\left(d_{i},T\left(d_{i}\right),E\left(d_{i}\right)\right)$, where $d_{i}$ denotes a specific disease, $T(d_{i})$ denotes the set containing the disease node $d_{i}$ and all its ancestor nodes, and $E(d_{i})$ denotes the set of corresponding edges. According to the constructed directed acyclic graph of disease $d_{i}$, we can calculate the semantic contribution value of disease $d_{k}$ to disease $d_{i}$ as follows:
(2)$$D_{d_{i}}^{1}\left(d_{k}\right)=\left\{\begin{array}{cc} 1, & \;\mathrm{if}\;d_{k}=d_{i}\\ \max\left\{\delta *D_{d_{i}}^{1}\left(d^{\prime }\right)|d^{\prime }\in \;\mathrm{children}\;\mathrm{of}\;d_{k}\right\}, & \;\mathrm{if}\;d_{k}\neq d_{i} \end{array}\right.,$$
where δ is the semantic contribution decay factor and based on a previous study, we set δ to 0.5. We can then calculate the semantic value of the disease $d_{i}$.
(3)$$DV^{1}\left(d_{i}\right)=\sum_{d_{k}\in T\left(d_{i}\right)}D_{d_{i}}^{1}\left(d_{k}\right).$$

Based on the assumption that the more the overlapping parts of the DAGs of two diseases are, the more similar they are. We can calculate the disease semantic similarity between diseases $d_{i}$ and $d_{j}$, and define it as follows:
(4)$$DS^{1}\left(d_{i},d_{j}\right)=\frac{\sum_{d^{\prime }\in T\left(d_{i}\right)\cap T\left(d_{j}\right)}\left(D_{d_{i}}^{1}\left(d^{\prime }\right)+D_{d_{j}}^{1}\left(d^{\prime }\right)\right)}{DV^{1}\left(d_{i}\right)+DV^{1}\left(d_{j}\right)},$$
where $DS^{1}$ is for storing the semantic similarity of the first kind of diseases.

However, the above calculation method has a disadvantage in that it does not account for the different contributions of two diseases in the same layer of the DAG, and the disease with a low frequency should contribute more than the disease with a high frequency. As a result, we developed a second semantic similarity model. Specifically, we can calculate the semantic contribution value of disease $d_{k}$ to disease $d_{i}$ as follows:
(5)$$D_{d_{i}}^{2}=-\log\left(\frac{\;\mathrm{the}\;\mathrm{number}\;\mathrm{of}\;\mathrm{DAGs}\;\mathrm{including}\;d_{k}}{\;\mathrm{the}\;\mathrm{number}\;\mathrm{of}\;\mathrm{diseases}\;}\right).$$

Likewise, we can obtain the semantic value of disease $d_{i}$:
(6)$$DV^{2}\left(d_{i}\right)=\sum_{d_{k}\in T\left(d_{i}\right)}D_{d_{i}}^{2}\left(d_{k}\right).$$

Based on the previously mentioned assumptions, we can calculate the second kind of disease semantic similarity between diseases $d_{i}$ and $d_{j}$, which is defined as follows: (7)$$DS^{2}\left(d_{i},d_{j}\right)=\frac{\sum_{d^{\prime }\in T\left(d_{i}\right)\cap T\left(d_{j}\right)}\left(D_{d_{i}}^{2}\left(d^{\prime }\right)+D_{d_{j}}^{2}\left(d^{\prime }\right)\right)}{DV^{2}\left(d_{i}\right)+DV^{2}\left(d_{j}\right)},$$
where $DS^{2}$ is for storing the second kind of disease semantic similarity. To obtain the sound disease semantic similarity, we combined the two types of disease semantic similarity to obtain the final disease semantic similarity, and the final disease semantic similarity between diseases $d_{i}$ and $d_{j}$ can be calculated according to the following equation:
(8)$$\mathrm{DSS}\left(d_{i},d_{j}\right)=\frac{DS^{1}\left(d_{i},d_{j}\right)+DS^{1}\left(d_{i},d_{j}\right)}{2}.$$

#### 2.2.3. Gaussian Interaction Profile Kernel Similarity of miRNAs and Diseases

Inspired by past studies [38], based on the hypothesis that functionally similar miRNAs may be associated with phenotypically similar diseases. We used Gaussian spectral kernel similarity to calculate the similarity between each pair of miRNAs and between each pair of diseases, which in turn complements the similarity information of miRNAs and diseases. Specifically, the Gaussian interaction profile kernel similarity between miRNAs $m_{i}$ and $m_{j}$ was calculated as follows: (9)$$\mathrm{GMS}\left(m_{i},m_{j}\right)=\exp\left(-\gamma_{m}{∥IP\left(m_{i}\right)-IP\left(m_{j}\right)∥}^{2}\right),$$
(10)$$\gamma_{m}=\gamma_{m}^{\prime }/\left(\frac{1}{n_{m}}\sum_{i=1}^{n_{m}}{∥IP\left(m_{i}\right)∥}^{2}\right),$$
where the parameter $\gamma_{m}$ controls the kernel bandwidth, which can be obtained based on the hyperparameter $\gamma_{m}^{\prime }$ normalized by the average number of interactions for each miRNA. According to previous studies, $\gamma_{m}^{\prime }$ is set to 1. For diseases, similar to miRNAs, the Gaussian interaction profile kernel similarity between diseases $d_{i}$ and $d_{j}$ is calculated as follows:
(11)$$\mathrm{GDS}\left(d_{i},d_{j}\right)=\exp\left(-\gamma_{d}{∥IP\left(d_{i}\right)-IP\left(d_{j}\right)∥}^{2}\right),$$
(12)$$\gamma_{d}=\gamma_{d}^{\prime }/\left(\frac{1}{n_{d}}\sum_{i=1}^{n_{d}}{∥IP\left(d_{i}\right)∥}^{2}\right),$$
where,$\gamma_{d}^{\prime }$ is set to 1.

#### 2.2.4. Integration of miRNAs and Diseases Similarity

Considering that some miRNAs have no function similarity to each other, and similarly, some diseases have no semantic similarity to each other, this can lead to a large number of sparse values in the miRNA function similarity matrix and disease semantic similarity matrix. To solve the above problem, we define the integrated similarity between miRNA $m_{i}$ and $m_{j}$ and the integrated similarity between diseases $d_{i}$ and $d_{j}$ by integrating the Gaussian interaction profile kernel similarity obtained from prior calculations as follows:
(13)$$S_{m}\left(m_{i},m_{j}\right)=\left\{\begin{array}{cc} MFS\left(m_{i},m_{j}\right), & \;\mathrm{if}\;m_{i}\;\mathrm{and}\;m_{j}\;\mathrm{have}\;\mathrm{function}\;\mathrm{similarity}\;\\ GMS\left(m_{i},m_{j}\right), & \;\mathrm{otherwise}\; \end{array}\right.,$$
(14)$$S_{d}\left(d_{i},d_{j}\right)=\left\{\begin{array}{cc} DSS\left(d_{i},d_{j}\right), & \;\mathrm{if}\;d_{i}\;\mathrm{and}\;d_{j}\;\mathrm{have}\;\mathrm{semantic}\;\mathrm{similarity}\;\\ GDS\left(d_{i},d_{j}\right), & \;\mathrm{otherwise}\; \end{array}\right..$$

### 2.3. SGAEMDA

To predict the potential association of miRNAs with diseases, we propose the stacked graph autoencoder miRNA–disease association prediction model (SGAEMDA) in this study. To successfully extract potential information in the similarity network and forecast miRNA–disease associations, the model integrates a graph convolutional network-based autoencoder with a multilayer perceptron. SGAEMDA is typically comprised of the following steps: (1) Construct initial features. (2) Pre-train stacked graph autoencoder to extract miRNA and disease similarity potential features. (3) Concatenate potential features and association features. (4) Predict miRNA-disease.

(1)Construct initial features

We construct the initial features of miRNAs and diseases from two different perspectives: Association information and similarity information. First, for the miRNA-disease association matrix A, each row can be regarded as the association feature of miRNA and each column as the association feature of disease. For the miRNA integrative similarity matrix $S_{m}$ and the disease integrative similarity matrix $S_{d}$, each row of $S_{m}$ can be regarded as the similarity feature of miRNA, and each row of $S_{d}$ can be regarded as the similarity feature of disease. Specifically, the two initial feature vectors of miRNAs and diseases are shown as follows: (15)$$F_{\phi }^{\ell }=\left(v_{1},v_{2},v_{3},\ldots ,v_{n_{\phi \ell }}\right),$$
where ℓ∈{1,2}, when ℓ=1, $F_{\phi }^{1}$ denotes the association feature of miRNA or disease, and when ℓ=2, $F_{\phi }^{2}$ denotes the functional similarity feature of miRNA or semantic similarity feature of disease. ϕ∈{m,d},ϕ=m represents miRNA features and ϕ=d represents disease features, and $n_{m1},n_{d1},n_{m2},n_{d2}$ are the number of columns and rows of *A*, the number of columns of $S_{m}$, and the number of columns of $S_{d}$, i.e., 383, 495, 495, and 383, respectively.

(2)Pre-train stacked graph autoencoder

Referring to a previous study [39], graph autoencoder can learn the low-dimensional feature representation of graph nodes to find the appropriate embedding. Since the information in the similarity features of miRNAs and diseases is high-dimensional, this could affect the prediction accuracy of the prediction model. We propose the stacked graph autoencoder to extract the low-dimensional similarity potential features from it, which has a stronger feature extraction ability than the traditional graph autoencoder. The graph autoencoder is particularly suitable for datasets with large numbers of unlabeled data and small numbers of labeled data due to its unsupervised training method. Specifically, the encoder and decoder for each layer of the autoencoder are defined as follows:
(16)$$\mathrm{Enc}\left(A,Y\right)=\tanh\left(A\cdot \mathrm{ReLU}\left(AYW^{0}\right)W^{1}\right),$$
and
(17)$$\mathrm{Dec}\left(A,Y\right)=\mathrm{sigmoid}\left(A\cdot \mathrm{ReLU}\left(AYW^{2}\right)W^{3}\right),$$
where A,Y,W denote the adjacency matrix, feature matrix of the node, and the learnable parameter matrix. Therefore, the feature representation of miRNA, $Z_{m}^{l}$ can be learned by the above encoder–decoder structure as follows:
(18)$$Z_{m}^{l}={\mathrm{Enc}}_{m}\left(A_{m},Z_{m}^{l-1}\right),$$
and
(19)$$X_{m}^{l}={\mathrm{Dec}}_{m}\left(A_{m},Z_{m}^{l}\right),$$
where *l* denotes the number of layers of the graph autoencoder, l=1,2…L, $Z_{m}^{l}$ denotes the low-dimensional feature representation learned by the lth layer of the graph autoencoder, when 1=1, $Z_{m}^{0}$, i.e., $F_{m}^{2}$, $X_{m}^{l}$ denotes the miRNA feature representation reconstructed by the lth layer of the autoencoder, and $A_{m}$ denotes the Laplace-normalized miRNA adjacency matrix. The formula is as follows:
(20)$$A_{m}=D_{m}^{-1/2}S_{m}D_{m}^{-1/2},$$
where $D_{m}$ is the degree matrix of miRNA-integrated similarity matrix $S_{m}$.

Similarly, we learn the low-dimensional feature representation $Z_{d}^{l}$ of the disease by the stacked graph autoencoder of the same architecture as follows: (21)$$Z_{d}^{l}={\mathrm{Enc}}_{d}\left(A_{d},Z_{d}^{l-1}\right),$$
and
(22)$$X_{d}^{l}={\mathrm{Dec}}_{d}\left(A_{d},Z_{d}^{l}\right),$$
where $Z_{d}^{l}$ denotes the low-dimensional feature representation learned by the lth layer graph autoencoder, when l=1, $Z_{d}^{0}$, i.e., $F_{d}^{2}$, $X_{d}^{l}$ denotes the disease feature representation reconstructed by the lth autoencoder, and $A_{d}$ denotes the Laplace-normalized adjacency matrix of the disease. The formula is as follows: (23)$$A_{d}=D_{d}^{-1/2}S_{d}D_{d}^{-1/2}.$$

In this study, SGAE is constructed by stacking three graph autoencoders, i.e., L=3. Specifically, the feature representation generated by the first graph autoencoder is taken as input to the second autoencoder, which generates another feature representation of lower dimensionality, and so on, until L graph autoencoders are constructed. Multiple graph autoencoders are trained collaboratively based on the reconstruction loss function to generate the final low-dimensional similarity feature representations of miRNA and disease, $Z_{m}^{L}$ and $Z_{d}^{L}$, with the following equations: (24)$${\mathrm{Loss}}_{m}=\sum_{l=1}^{L}{∥Z_{m}^{l-1}-X_{m}^{l}∥}^{2},$$
(25)$${\mathrm{Loss}}_{d}=\sum_{l=1}^{L}{∥Z_{d}^{l-1}-X_{d}^{l}∥}^{2}.$$

(3)Concatenate potential features and association features

We set the final embedding dimension to 64 in pre-training, and the training obtained a low-dimensional similarity representation of all miRNAs and diseases, denoted as $Z_{m}^{L}$, $Z_{d}^{L}$, respectively. To include more potential information in the feature representations of miRNAs and diseases, we concatenated $Z_{m}^{L}$ and $Z_{d}^{L}$ with the association feature $F_{m}^{1}$ of miRNAs and the association feature $F_{d}^{1}$ of diseases, respectively, and finally obtained a 447-dimensional miRNA embedding and a 559-dimensional disease embedding, as follows: (26)$$V_{m}=\;\mathrm{concatenating}\;\left(Z_{m}^{L},F_{m}^{1}\right),$$
and
(27)$$V_{d}=\;\mathrm{concatenating}\;\left(Z_{d}^{L},F_{d}^{1}\right),$$
where $V_{m}$ denotes the final embedding of miRNA and $V_{d}$ denotes the final embedding of disease.

(4)Predict miRNA-disease association by multilayer perceptron

After obtaining the embedding of miRNAs and diseases, we concatenate the embedding $V_{m_{i}}$ for each miRNA and $V_{d_{j}}$ for each disease to form our complete dataset X, where X∈$R^{\left(495*383)\times (447+559\right)}$, as follows:
(28)$$X_{ij}=\;\mathrm{concatenating}\;\left(V_{m_{i}},V_{d_{j}}\right),$$
where $X_{ij}$ denotes the characteristics of miRNA-disease pairs of miRNA $m_{i}$ and disease $d_{j}$. Then, we used a multilayer perceptron (MLP) to score the final miRNA-disease association for prediction, as follows:
(29)$$X^{l}=\mathrm{ReLU}\left(X^{l-1}W^{l}+b^{l}\right),$$
and
(30)$${\hat{y}}_{ij}=\mathrm{Sigmoid}\left(X^{2}W^{3}+b^{3}\right),$$
where l∈[1,2] denotes the number of layers of the hidden layer, $X^{l}$ denotes the output of the lth hidden layer, and $W^{l},b^{l}$ are the learnable parameter matrix and bias of the lth hidden layer, respectively. ${\hat{y}}_{ij}$ is the prediction score of the final miRNA-disease pair. Finally, the model is trained by minimizing the error of the Binary Cross-Entropy Loss function:
(31)$$Loss=-\frac{1}{N}\left(\sum_{\left(i,j\right)\in y^{+}}y_{ij}\log{\hat{y}}_{ij}+\sum_{\left(i,j\right)\in y^{-}}\left(1-y_{ij}\right)\log\left(1-{\hat{y}}_{ij}\right)\right),$$
where (i,j) denotes the pair for miRNA $m_{i}$ and disease $d_{j}$.$y^{+}$ and $y^{-}$ subtables denote the positive and negative sample sets. N denotes the number of all miRNA-disease pairs in the positive and negative sample sets.

## 3. Results

### 3.1. Experiment Details

In our experiments, the SGAEMDA model is implemented based on the pytorch framework and the scikit-learn framework. The Adam optimizer is adopted to minimize the loss function both during the pre-training process and the MLP training process. Due to the significant imbalance of positive and negative samples in the database of HMDDv2.0, the number of known miRNA–disease associations is 5430 (positive samples), and the rest of the 184,155 pairs are unknown associations (negative samples), and the number of negative samples is about 34 times the positive samples. In order to have good robustness of our model, we randomly selected negative samples equal to the positive samples for MLP training, and randomly selected 10 times in the subsequent experiments to ensure the reliability of our experiments. Our source code of HSSG is available online: https://github.com/Lynn0424/SGAEMDA (accessed on 5 December 2022).

### 3.2. Evaluation Metrics

The area under the receiver operating characteristic curve (AUC) and area under precision–recall curve (AUPR) were our main metrics to evaluate the overall model performance. In classification problems, AUC is an essential method to evaluate the overall performance of a model, and for unbalanced data sets, AUPR can evaluate the model better than AUC. In order to be more comprehensive in evaluating the performance of the SGAEMDA model, we also used several common evaluation metrics such as accuracy (Acc), precision (Pre), recall (Rec), and F1-score. Several metrics are calculated as follows: (32)$$\;\mathrm{Acc}\;=\frac{TP+TN}{TP+TN+FP+FN},$$
(33)$$\;\mathrm{Pre}\;=\frac{TP}{TP+FP},$$
(34)$$\;\mathrm{Rec}\;=\frac{TP}{TP+FN},$$
(35)$$\;F1-\mathrm{score}\;=\frac{2\times \;\mathrm{Pre}\times \mathrm{Rec}}{\;\mathrm{Pre}\;+\;\mathrm{Rec}\;},$$
where TP, TN, FP, FN denote true positive, false negative, false positive, and true negative, respectively.

### 3.3. Prediction of miRNA–Disease Association Based on SGAEMDA

To obtain reliable experimental results of the model, we performed 5-fold cross-validation and 10-fold cross-validation to evaluate the model performance of SGAEMDA. In 5-fold CV (10-fold CV), all the training samples are randomly divided into 5 (10) subsets of approximately the same number, 4 (9) of them are chosen for training and the remaining 1 is chosen for testing, and the process is repeated until all the subsets have been used for the test set, and finally the obtained results are averaged as the final result. Figure 2 and Figure 3 show the ROC curves and PR curves for the 5-fold CV and 10-fold CV and the area under their curves. It can be seen that our model has an AUC above 0.95 for both 5-fold CV and 10-fold CV, indicating the effectiveness of the model in predicting the potential miRNA-disease association and implying that the model performance is not affected by the amount of training data and test data in cross-validation. Table 1 shows the average results of other evaluation metrics and their standard deviations for 5-fold CV and 10-fold CV, indicating the ACC, Pre, Rec, F1-score of SGAEMDA at 5-fold CV (10-fold CV) of 0.9045 (0.9087), 0.9037 (0.8949), 0.9056 (0.9272), 0.9046 (0.9104). The SGAEMDA model was further demonstrated to be effective for association prediction.

### 3.4. Effect of Similarity Feature Dimensions

To further illustrate the effect of the final dimensionality of the similarity features on the model prediction performance, we set the dimensionality of the similarity features learned by the stacked graph autoencoder to 16, 32, 64, 128, 256 for comparison experiments, and calculate their AUC and AUPR, respectively. The experimental results are shown in Figure 4, and both AUC and AUPR reach the highest value when the dimension is 64. Therefore, we set the final learned similarity feature dimension to 64. In addition, we can infer that if the dimension is too small, it cannot fully learn the similarity information; while if the dimension is too large, there may be original redundant and noisy information, leading to lower model performance.

### 3.5. Effect of Stacked Graph Autoencoder Pre-Training

In SGAEMDA, to verify the validity of our proposed stacked graph autoencoder for miRNA–disease potential association prediction. We designed three groups of experiments. The first one uses only the potential similarity features $\left(Z_{m}^{L}\right.$ and $\left.Z_{d}^{L}\right)$ obtained by pre-training and uses them directly as the final embedding of miRNAs and diseases for prediction, denoted as only-pre-training. The second group is a direct concatenation of the original similarity features $\left(F_{m}^{2}\right.$ and $\left.F_{d}^{2}\right)$ and association features $\left(F_{m}^{1}\right.$ and $\left.F_{d}^{1}\right)$ for prediction without using stacked graph autoencoder, denoted as non-pre-training. The third group uses only the original association features $\left(F_{m}^{1}\right.$ and $\left.F_{d}^{1}\right)$ to predict the potential association, which is denoted as only-original feature. The fourth group of experiments uses pre-trained features $\left(Z_{m}^{L}\right.$ and $\left.Z_{d}^{L}\right)$ and association features $\left(F_{m}^{1}\right.$ and $\left.F_{d}^{1}\right)$, i.e., the SGAEMDA model.

Figure 5 and Table 2 show the prediction results of the four models. We can see that the SGAEMDA model is only slightly lower than the only-original feature model in Recall, but reaches the highest value in all the rest of the metrics. AUC and AUPR are more reflective of the overall performance of the model, so integrating the features learned by stacked autoencoder and association features can enable the model to achieve better performance.

### 3.6. Comparison of Different Classifier Models

In the SGAEMDA model, we used a multilayer perceptron (MLP) classifier to predict the potential miRNA–disease association. To confirm the reasonability of our adopted MLP, we used cross-validation with the same dataset for comparison with four common classifier models, which are random forest (RF), support vector machine (SVM), K-nearest neighbor (KNN), and XGBoost algorithm. We refer to the Liu et al. [21] proposed model to select the best parameters for different classifiers. In the RF algorithm, we set the maximum depth of the tree to 10, the maximum features to 100, and the rest of the parameters to default values. In the SVM algorithm, we use the RBF kernel and set C to 50. In the XGBoost algorithm, we set the number of trees to 1000, the learning rate to 0.1, and the rest of the parameters to their default values. For the KNN classifier, we performed a parameter sensitivity analysis and finally set the K value to 4, the *p*-value to 2, and the rest of the parameters to their default values. Table 3 shows the prediction performance of these classifiers. It can be seen that SGAEMDA achieves the highest results in four of the five evaluated metrics, and only in the accuracy rate it is 2.07% lower than the KNN classifier. However, in terms of potential association prediction, AUC and AUPR are more likely to show the overall model performance. Therefore, we selected MLP as our final classifier.

### 3.7. Comparisons with Existing SOTA Methods

To further prove the predictive performance of our proposed SGAEMDA model, we compare it with nine state-of-the-art existing computational models, namely LAGCN [28], GBDT-LR [19], EKRRMDA [20], MCLPMDA [29], GAEMDA [25], PMDFI [22], SMALF [21], DAEMKL [26], and DFELMDA [23]. Since the AUC values provide a comprehensive measure of the overall predictive performance of the models, we selected the AUC as a metric to evaluate the performance of these models (all AUC values were selected from their papers by taking their best values). In addition, the above models are all evaluated based on HMDDv2.0 on the five-fold cross-validation basis. Table 4 shows the comparative results of the models. From the table, we see that SGAEMDA achieved the highest AUC value among the 10 models, which is 3.3% higher than the second-best model (DFELMDA). In conclusion, SGAEMDA has very good results in predicting potential miRNA–disease associations.

### 3.8. Case Studies

We selected four neoplastic diseases as case studies: brain neoplasms (Table 5), breast neoplasms (Table 6), colon neoplasms (Table 7), and kidney neoplasms (Table 8). Specifically, there are 5430 known miRNA-disease associations in the HMDDv2.0 database, while the remaining 184,155 associations are unknown. The known associations obtained from the database were used as the training set for SGAEMDA, and then we prioritized the candidate miRNAs for several neoplasms based on the prediction scores and selected the top 20 candidate miRNAs. We verified the predicted experimental results one by one by using the dbDEMCv3.0 database [40] and the miRCancer database [41] as validation sets.

Brain neoplasms are defined as a neoplasm growing in the cranial cavity, also known as brain cancer and intracranial neoplasm. They are generally divided into two categories: primary and secondary [42]. Statistics show that the incidence of brain neoplasm has been increasing in recent years, and brain neoplasm accounts for about 5% of the total body neoplasms, other malignant neoplasms in the body have a 20–30% probability to metastasize into the skull, once the neoplasm occupies a certain space in the skull, regardless of benign or malignant neoplasm, it will endanger the life of patients. According to statistics, the incidence of brain neoplasms has been increasing in recent years. Brain neoplasms account for about 5% of the whole-body neoplasms, and all other malignant tumors in the body have a 20–30% chance of metastasizing to the skull. Therefore, a research priority was given to investigate miRNAs that may be associated with brain cancer. The results are shown in Table 5. Among the top 20 miRNAs associated with brain cancer, 19 of them are confirmed by dbDEMC or miRCancer.

It is estimated that breast neoplasms account for 7–10% of all malignant tumors in the body. Its incidence is generally associated with genetics and is higher in women between 40–60 years of age [43,44]. Thus, the discovery of potential miRNAs associated with breast neoplasms provides direction for the treatment and diagnosis of breast neoplasms. The results are shown in Table 6 and all 20 of the predicted miRNAs associated with breast cancer are confirmed.

Colon cancer, also known as colorectal cancer, is a malignant neoplasm of the gastrointestinal tract that occurs in the colon area. The incidence of colon neoplasms is statistically second only to gastric and esophageal cancers [45]. As shown in Table 7, it can be seen that 19 of the top 20 miRNAs predicted to be potentially associated with colon cancer are confirmed.

Kidney neoplasms have a high incidence in western countries [46]. In addition, about 95% of renal neoplasms are malignant, the pathology of kidney tumors is more complex, and it is more challenging to treat kidney tumors. Table 8 shows that 19 of the top 20 miRNAs were validated by the database.

In addition, to further validate the performance of our model, we downloaded miRNAseq data for BRCA (breast invasive carcinoma) and COADREAD (colorectal cancer) from the TCGA database. Based on the downloaded data, we compared the differential expression between the top 10 miRNA paracancer sample groups that we predicted. The results of differential expression are shown in Figure 6.

## 4. Discussion

In the past, many studies have shown that aberrant miRNA expression is often associated with many biological processes as well as the occurrence of complex diseases in humans with considerable impact. Thus, predicting potential miRNA-disease associations can help medical professionals provide molecular insight into the pathogenesis of various complex diseases and thus develop relevant new drugs. In this paper, we propose the SGAEMDA model, a novel model based on a stacked graph autoencoder. Unlike previous stacked-autoencoders, SGAE is not trained layer-by-layer but in collaboration with each layer, which makes up for the drawback of weak coding ability due to greedy training of previous stacked-autoencoders. It can extract potential feature representations from miRNA similarity networks and disease similarity networks at a deeper level. The extracted features are concatenated with the corresponding association features and then MLP is used to predict the association between miRNA and diseases. After experiments, it is shown that the highest AUC value of SGAEMDA, which reached 0.9585 under the 5-fold and 10-fold cross-validation. is much higher than the other baseline methods. The case study analysis experimentally confirmed that our model can effectively predict the potential miRNA-disease association. However, our work still has some areas for improvement:
(1)The model is not trained end-to-end, and our model may be lower in robustness.(2)The data used in the experiments are fewer and unable to extract more information about miRNAs and diseases from more perspectives.

In future studies, we will fuse more miRNA and disease similarity information to further improve the performance of our prediction models. Moreover, we will utilize a scheme similar to the EGES model [47] to allow embedding to cover more miRANs and diseases, thus addressing the cold-start problem in genetic disease association prediction.

## Figures and Tables

**Figure 2 cells-11-03984-f002:**
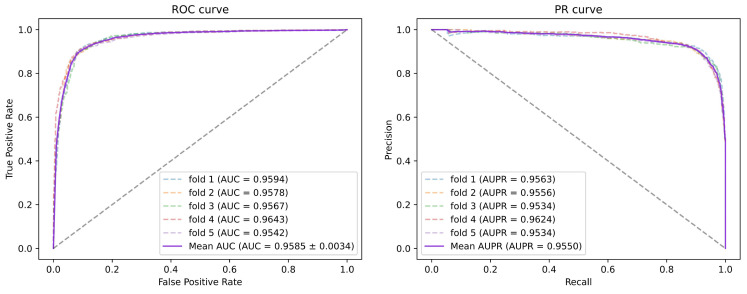
The 5-fold cross-validated ROC curve and PR curve of SGAEMDA model with AUC of 95.85% and AUPR of 95.50%.

**Figure 3 cells-11-03984-f003:**
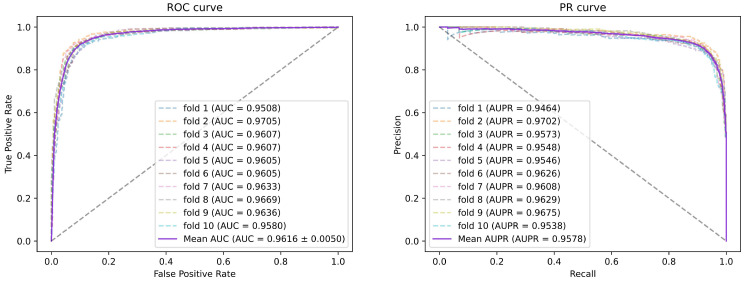
The 10-fold cross-validated ROC curve and PR curve of SGAEMDA model with 96.16% AUC and 95.78% AUPR.

**Figure 4 cells-11-03984-f004:**
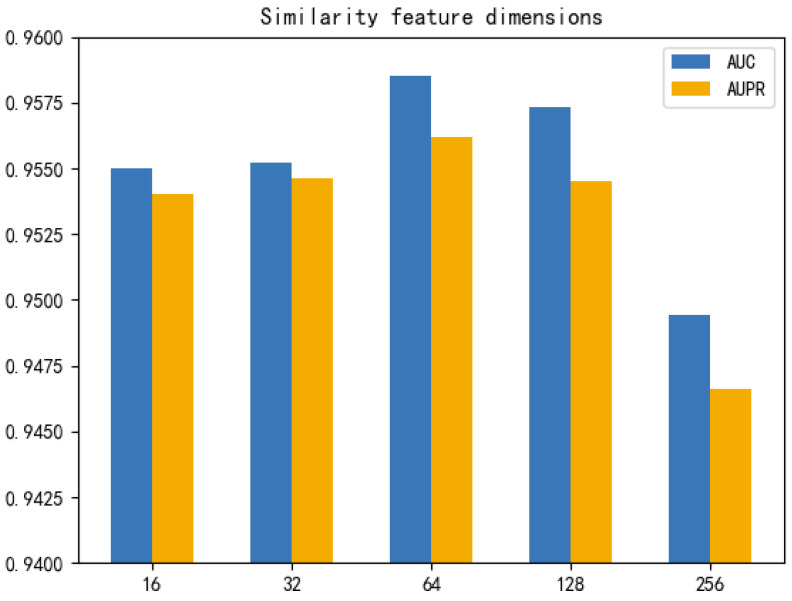
AUC and AUPR in different similarity feature dimensions under 5-fold CV.

**Figure 5 cells-11-03984-f005:**
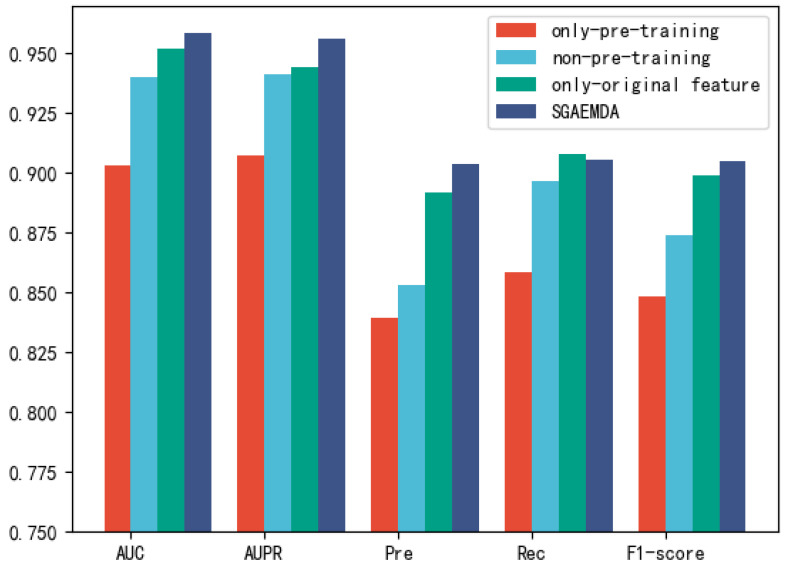
Comparison of the prediction effect of different models.

**Figure 6 cells-11-03984-f006:**
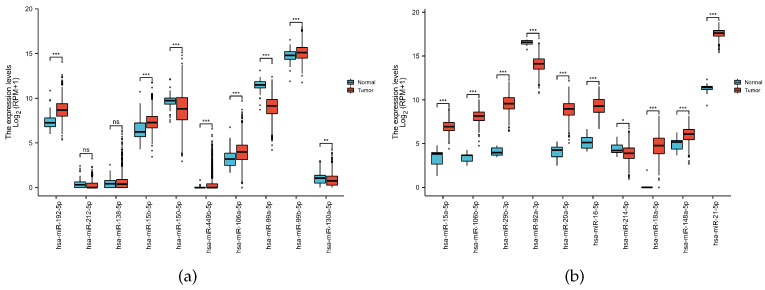
The result of the miRNA differential expression. (**a**) miRNAs ranked 1–10 for breast cancer. (**b**) miRNAs ranked 1–10 for colon cancer.

**Table 1 cells-11-03984-t001:** The 5-fold and 10-fold cross-validation results of the SGAEMDA model.

Cross- Validation	Acc	Pre	Rec	F1-Score
5-fold CV	0.9045 ± 0.003	0.9037 ± 0.008	0.9056 ± 0.010	0.9046 ± 0.004
10-fold CV	0.9087 ± 0.007	0.8949 ± 0.022	0.9272 ± 0.016	0.9104 ± 0.006

**Table 2 cells-11-03984-t002:** Comparison table of each evaluation metric for different models.

	AUC	AUPR	Pre	Rec	F1-Score
only-pre-training	0.9031	0.9071	0.8394	0.8582	0.8486
non-pre-training	0.9402	0.9409	0.8530	0.8967	0.8739
only-original feature	0.9422	0.9442	0.8920	0.9076	0.899
SGAEMDA	0.9585	0.9562	0.9037	0.9056	0.9046

**Table 3 cells-11-03984-t003:** Five types of classifier evaluation metrics.

	AUC	AUPR	Pre	Rec	F1-Score
RF	0.9356	0.9351	0.8505	0.872	0.8611
SVM	0.934	0.933	0.8601	0.8506	0.8553
KNN	0.9282	0.9399	0.9244	0.7703	0.8401
XGBoost	0.9538	0.9545	0.8876	0.8833	0.8854
SGAEMDA	0.9585	0.9562	0.9037	0.9056	0.9046

**Table 4 cells-11-03984-t004:** Comparison of different methods based on 5-fold cross-validation.

Method	AUC(%)
LAGCN	90.91
GBDT-LR	92.74
EKRRMDA	92.75
MCLPMDA	93.20
GAEMDA	93.56
PMDFI	94.04
SMALF	95.03
DAEMKL	95.38
DFELMDA	95.52
SGAEMDA	95.85

**Table 5 cells-11-03984-t005:** Top 20 brain neoplasm-related miRNAs predicted by SGAEMDA based on HMDD v2.0.

TOP 1-10 miRNA	dbDEMC	miRCancer	TOP 11-20 miRNA	dbDEMC	miRCancer
hsa-mir-221	Comfirmed	Comfirmed	hsa-mir-101	Comfirmed	Uncomfirmed
hsa-mir-26b	Comfirmed	Uncomfirmed	hsa-mir-184	Comfirmed	Uncomfirmed
hsa-mir-106b	Comfirmed	Uncomfirmed	hsa-mir-218	Comfirmed	Uncomfirmed
hsa-mir-181a	Comfirmed	Uncomfirmed	hsa-mir-146a	Comfirmed	Uncomfirmed
hsa-mir-155	Comfirmed	Uncomfirmed	hsa-mir-302b	Comfirmed	Uncomfirmed
hsa-mir-148a	Comfirmed	Uncomfirmed	hsa-mir-206	Comfirmed	Uncomfirmed
hsa-mir-125b	Comfirmed	Uncomfirmed	hsa-mir-197	Comfirmed	Uncomfirmed
hsa-mir-195	Comfirmed	Uncomfirmed	hsa-mir-196a	Comfirmed	Uncomfirmed
hsa-mir-210	Comfirmed	Uncomfirmed	hsa-mir-410	Comfirmed	Uncomfirmed
hsa-mir-200c	Uncomfirmed	Uncomfirmed	hsa-mir-214	Comfirmed	Uncomfirmed

**Table 6 cells-11-03984-t006:** Top 20 breast neoplasm-related miRNAs predicted by SGAEMDA based on HMDD v2.0.

TOP 1-10 miRNA	dbDEMC	miRCancer	TOP 11-20 miRNA	dbDEMC	miRCancer
hsa-mir-192	Comfirmed	Uncomfirmed	hsa-mir-144	Comfirmed	Comfirmed
hsa-mir-212	Comfirmed	Comfirmed	hsa-mir-185	Comfirmed	Comfirmed
hsa-mir-138	Comfirmed	Comfirmed	hsa-mir-449a	Comfirmed	Comfirmed
hsa-mir-15b	Comfirmed	Uncomfirmed	hsa-mir-98	Comfirmed	Comfirmed
hsa-mir-150	Comfirmed	Comfirmed	hsa-mir-542	Comfirmed	Uncomfirmed
hsa-mir-449b	Comfirmed	Comfirmed	hsa-mir-424	Comfirmed	Uncomfirmed
hsa-mir-106a	Comfirmed	Comfirmed	hsa-mir-92b	Comfirmed	Uncomfirmed
hsa-mir-99a	Comfirmed	Comfirmed	hsa-mir-181d	Comfirmed	Uncomfirmed
hsa-mir-99b	Comfirmed	Uncomfirmed	hsa-mir-186	Comfirmed	Comfirmed
hsa-mir-130a	Comfirmed	Comfirmed	hsa-mir-376a	Comfirmed	Comfirmed

**Table 7 cells-11-03984-t007:** Top 20 colon neoplasm-related miRNAs predicted by SGAEMDA based on HMDD v2.0.

TOP 1-10 miRNA	dbDEMC	miRCancer	TOP 11-20 miRNA	dbDEMC	miRCancer
hsa-mir-15a	Comfirmed	Comfirmed	hsa-mir-19b	Comfirmed	Comfirmed
hsa-mir-106b	Comfirmed	Uncomfirmed	hsa-mir-195	Comfirmed	Comfirmed
hsa-mir-29b	Comfirmed	Uncomfirmed	hsa-mir-122	Comfirmed	Uncomfirmed
hsa-mir-92a	Comfirmed	Uncomfirmed	hsa-mir-26a	Uncomfirmed	Uncomfirmed
hsa-mir-20a	Comfirmed	Comfirmed	hsa-mir-125a	Comfirmed	Comfirmed
hsa-mir-16	Uncomfirmed	Comfirmed	hsa-mir-93	Comfirmed	Comfirmed
hsa-mir-214	Comfirmed	Comfirmed	hsa-mir-141	Comfirmed	Comfirmed
hsa-mir-18a	Comfirmed	Comfirmed	hsa-mir-20b	Comfirmed	Uncomfirmed
hsa-mir-148a	Comfirmed	Uncomfirmed	hsa-mir-10a	Comfirmed	Uncomfirmed
hsa-mir-21	Comfirmed	Comfirmed	hsa-mir-30b	Comfirmed	Uncomfirmed

**Table 8 cells-11-03984-t008:** Top 20 kidney neoplasm-related miRNAs predicted by SGAEMDA based on HMDD v2.0.

TOP 1-10 miRNA	dbDEMC	miRCancer	TOP 11-20 miRNA	dbDEMC	miRCancer
hsa-mir-145	Comfirmed	Comfirmed	hsa-mir-200b	Comfirmed	Uncomfirmed
hsa-mir-29b	Comfirmed	Uncomfirmed	hsa-mir-126	Comfirmed	Uncomfirmed
hsa-mir-214	Comfirmed	Uncomfirmed	hsa-mir-210	Comfirmed	Comfirmed
hsa-mir-106b	Comfirmed	Uncomfirmed	hsa-mir-195	Comfirmed	Uncomfirmed
hsa-mir-122	Comfirmed	Uncomfirmed	hsa-mir-23a	Comfirmed	Uncomfirmed
hsa-mir-15b	Comfirmed	Uncomfirmed	hsa-mir-155	Comfirmed	Uncomfirmed
hsa-mir-106a	Comfirmed	Uncomfirmed	hsa-mir-375	Comfirmed	Comfirmed
hsa-mir-143	Comfirmed	Uncomfirmed	hsa-mir-31	Comfirmed	Uncomfirmed
hsa-mir-1	Uncomfirmed	Uncomfirmed	hsa-mir-223	Comfirmed	Comfirmed
hsa-mir-429	Comfirmed	Uncomfirmed	hsa-mir-212	Comfirmed	Uncomfirmed

## Data Availability

Known miRNA–disease association data were taken from database HMDD 2.0 (http://www.cuilab.cn/hmdd, accessed on 23 May 2022), human microRNA functional similarity (http://www.lirmed.com/misim/, accessed on 23 May 2022), and disease semantic similarity (https://www.nlm.nih.gov/mesh/, accessed on 23 May 2022).

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
