# Peer review of "SGAEMDA: Predicting miRNA-Disease Associations Based on Stacked Graph Autoencoder"

_cells, 2022, doi:10.3390/cells11243984_

Round 1
Reviewer 1 Report
The authors introduced some computational methods for predicting potential miRNA-disease associations. They proposed a multi-layer collaborative unsupervised training base model called SGAEMDA. They adopted the sequence information of miRNAs, the MeSH information of diseases, and the known MDAs. They used the known MDAs to construct the gaussian interaction profile kernel similarity and fused multi-source information. Then, they used the stacked graph autoencoder to extract the low dimensional feature and adopted MLP to classify the potential MDAs. In general, the experimental work is sufficient. The topic is creative and meaningful. However, there are some major issues with the manuscript which need to be dealt with before this manuscript is accepted.
1. MDAs network is sparse, how can you handle this situation?
2. Try to test the proposed model in different scenarios (e.g., new miRNA vs. new disease, similar to cold-start issue)
3. Figure 1, and 2 are too simplistic, the authors are supposed to go further to illustrate the information in them.
4. The authors are suggested specify the meaning of MFS(·,·), GMS(·,·) in Equation 9 and GDS(·,·) in Equation 10.
5. If possible, the authors are supposed to make all data and software code on which the conclusions of the paper rely available to readers.
6. Your manuscript needs careful editing and particular attention to English grammar, spelling, and sentence structure.
Reviewer 2 Report
The study is interesting because the authors used a stacked graph autoencoder, joined with a MLP, to predict potential miRNAs and disease interactions. However, I have the following comments:
a) Figure 1 has some typos; for example, it says "Normorlization" which I believe should say "Normalization." Additionally, the sub-labels (1) to (4) are not depicted in the graph.
b) After eq. 4, what do the authors refer when they mention that two diseases share the same layer?
c) There is a typo in formula 6; it should read summation of D2.
d) The explanation of eq 11 requires to put on more details. For example, when it mentions the values of l, does this refers to association (similarity) information between miRNAs and diseases, or are they semantic or functional relationships? Additionally, it needs to be clarified what the values 383, 495, etc., represent.
e) Equation 12 would be needed to clarify what the input X refers to.
f) Before eq. 22, it would be valuable to mention the original values of Z before the concatenation.
g) The hyperparameters selected for the different ML methods compared in section 3.6 should be mentioned; also, it would be advisable to describe how they were obtained.
Reviewer 3 Report
Implementing computational models to explore miRNA-disease associations is one of the hot topics of research today. In particular, this paper presents a novel approach to predict miRNA-disease associations using graph autoencoder. Such a consideration is relatively novel and meaningful. The experiments seem to be good. The method shows better performance compared to the previous methods. However, there are still some major issues that should be of concern to the authors:
(1) Equation 9 and Equation 10 are not well formatted,
(2) Table5, 6, 7 and 8 chart formats need to be corrected,
(3) In order to apply the algorithm more widely, we suggest that the authors give links to the source code so that other researchers can conduct more in-depth research based on your model,
(4) Include more related published works as part of the literature review. Some of them are mentioned here, but not limited to those only.
https://doi.org/10.1186/s12859-022-04796-7
